# Analgesic Effectiveness of Physical Therapy Combining the Use of Electromagnetic Fields with Light Radiation Emitted by LEDs along with the Use of Topical Herbal Ointment in Patients with Gonarthrosis

**DOI:** 10.3390/ijerph20043696

**Published:** 2023-02-19

**Authors:** Katarzyna Janczewska, Kamil Koszela, Robert Klimkiewicz, Anna Kubsik-Gidlewska, Agnieszka Jankowska, Paulina Klimkiewicz, Marta Woldańska-Okońska

**Affiliations:** 1Department of Internal Medicine, Rehabilitation and Physical Medicine, Medical University, 90-419 Łódź, Poland; 2Neuroorthopedics and Neurology Clinic and Polyclinic, National Institute of Geriatrics, Rheumatology and Rehabilitation, 02-637 Warsaw, Poland

**Keywords:** knee osteoarthritis, magnetic therapy, conservative treatment

## Abstract

(1) Background: The aim of the study is to evaluate the analgesic effectiveness of a physical therapy regimen that combines the use of an electromagnetic field with light radiation emitted by LEDs, along with the use of Traumeel S ointment, in patients with gonarthrosis. (2) Methods: The study included 90 patients with knee osteoarthritis (grade 2 Kellgren and Lawrence osteoarthritis). They were divided into three groups: Group I, 30 patients treated with magnetic stimulation plus LED therapy; Group II, 30 patients treated with Traumeel S ointment; and Group III, 30 patients treated with magnetic stimulation plus LED therapy with Traumeel S ointment. Pain intensity was assessed using the VAS and Laitinen scales before and after a series of treatments. (3) Results: Significant results in terms of pain reduction before and after treatment were obtained in each of the study groups, as there were significant differences in the VAS pain intensity scores before and after the procedures between the groups. In group I, with electromagnetic field and LED light treatment, the difference was 35.5; in group II, which received Traumeel S^®^ ointment, the difference was 18.5; and in group III, with electromagnetic field and LED light treatment as well as Traumeel S ointment, the difference was 26.5. In the Laitinen scale, the differences were insignificant, although the size distribution was similar. (4) Conclusions: The therapy used in this study showed that magnetic stimulation plus LED therapy and the use of Traumeel S ointment gave positive results in terms of pain reduction in each of the study groups. The strongest analgesic factor seems to be magnetic and LED therapies used separately. Traumeel S in magnetoledophoresis does not work synergistically with the magnetic field of LED light, and even worsens the effect of the therapy used.

## 1. Introduction

Osteoarthritis (OA) is a slow, progressive, premature usage and degeneration of the tissues that make up the joints. OA is one of the most common diseases affecting the musculoskeletal system, which in turn leads to the degeneration of the articular cartilage, the subchondral layer of the bone, and as a result, it affects all layers forming the joint. Osteoarthritis of the knee joints (gonarthrosis, GA) is the third most frequently occurring form of osteoarthritis, behind osteoarthritis of the hip and spine joints [1].

The disease especially affects the elderly, and the incidence rate increases with age. It mainly affects people aged 65–74, which represents 10–20% of the world’s population, while after the age of 75, the incidence rate increases to 40% [1,2]. OA manifests itself in patients between 40 and 60 years of age and may initially be asymptomatic, although it may manifest itself earlier through molecular changes. Based on epidemiological data, it can be concluded that the disease affects women more often, causing its more severe forms [3]. Etiologically, OA is divided into primary and secondary OA [1,4,5]. The first clinical symptom of OA is pain that increases with physical activity and decreases with rest, followed by morning stiffness. As the disease progresses, the joint becomes inflamed, which may cause effusion and pain at rest and/or at night. Median CRP values significantly correlate with functional disability, pain, joint tenderness, overall severity of osteoarthritis, fatigue and associated depression. The average level of CRP in OA is higher than in healthy people. In the advanced stage of the disease, the contours of the joint become distorted and enlarged, and structural intra-articular changes also occur, which in turn impairs range of motion and general functioning in everyday life. The most common causes of secondary changes are disorders of the axis of the lower limb (walgus or varus of the knee joints), overload changes resulting from the assumed position (kneeling and squatting), and excessive body weight. The cause may be developmental disorders of the joint, e.g., knee dysplasia or changes of overload and traumatic nature [5,6].

Interestingly, obesity is also associated not only with osteoarthritis of the hips but also of the hands. This indicates that excess adipose tissue produces humoral factors, altering the metabolism of articular cartilage. It has been postulated that the leptin system may be a link between metabolic abnormalities in obesity and an increased risk of osteoarthritis [5].

Current diagnosis and treatment of knee osteoarthritis (KOA) is based mainly on clinical and imaging symptoms, ignoring its molecular pathophysiology. The mismatch between the molecular characteristics of the patients and the mechanisms of drug therapy may explain the failure of some disease-modifying drugs in clinical trials [6].

One of the most important elements in preventing the development of knee osteoarthritis is physiotherapy. In the fight against OA, the most effective treatment is rehabilitation that is well matched to the patient (age, health condition and comorbidities), taking into account both traditional kinesiotherapy and physical therapy. An important consideration is to relieve the knee joint by avoiding carrying heavy objects or wearing properly selected orthopedic equipment (insoles, shoes) and, above all, by controlling body weight [7,8].

Among the many treatments available, the most effective form of therapy to relieve pain and minimize degenerative progress in KOA is still being sought. In this context, it is interesting that plant extracts and their metabolites can affect diagnostic and prognostic KOA biomarkers [9,10].

Magnetic field LED therapy is common physical therapy modality with a high therapeutic efficacy, due to the combined effects of extremely low frequency electromagnetic fields (ELF-EMFs) and high-power light emitted by light-emitting diodes (LEDs) [11,12].

The importance of physical methods in therapy is constantly increasing due to their effectiveness, low cost, and the high degree of patient acceptance for this form of treatment. Magnetophoresis is a transdermal method (transdermal therapeutic system, TTS) for supporting drug penetration through biological barriers, which was used together with ketoprofen gel, achieving a positive analgesic effect that persisted even after 30 days (biological hysteresis phenomenon) [13]. The assumption of this study was to use an enhanced physical factor of magnetic stimulation plus LED therapy [14] in the form of magnetoledophoresis accompanied by the use of an ointment, with previously proven effectiveness [15].

Traumeel S is an ointment that works based on plant and mineral ingredients in a different way than do NSAIDs. It has been found that Traumeel S reduces the activity of NADPH oxidase and extracellular neutrophil traps, and also induces an anti-inflammatory effect on lymphocyte subpopulations [15].

The ointment relieves pain and swelling and reduces inflammation in the case of minor injuries. In terms of its composition, 100 g of ointment (Traumeel S^®^) contains 1.5 g Arnica montana; 0.45 g each calendula officinalis and Hamamelis virginiana; 0.15 g each Echinacea purpurea, Echinacea angustifolia, and Chamomilla recutita; 0.1 g each Symphytum officinale and Bellis perennis; 0.09 g each Hypericum perforatum and Achillea millefolium; 0.05 g each Aconitum napellus D1 and Dummy belladonna D1; 0.025 g Hepar sulfuris; and 0.04 g Mercurius solubilis Hahnemanni. The base of the ointment is a hydrophilic substance, DAB 10, preserved with 12.5% ethanol. Its effectiveness was determined in a study by González et al. in which it was compared with diclofenac. In rare cases, reactions associated with individual hypersensitivity may occur in the form of local allergic reactions (dermatitis, redness, swelling and itching). In this case, the drug should be discontinued and, if necessary, treatment of these symptoms should be initiated [16].

The aim of the study is to assess the efficiency of an electromagnetic field combined with light radiation emitted by LEDs and with Traumeel S^®^ ointment in the physiotherapy of patients with knee osteoarthritis according to the VAS and Laitinen scales.

## 2. Materials and Methods

### 2.1. Study Population

The study covered 90 patients, both sexes, with knee osteoarthritis (63 women and 27 men) aged 31–87 years. The research was conducted at the Rehabilitation and Physical Medicine Clinic of the Medical University of Lodz in the years 2013–2018. Patients were enrolled in the study after clinical and radiological examination, following exclusion and inclusion conditions. Each patient was assigned to a group based on the order of admission to the clinic.

The study was authorized by the Bioethics Committee for Scientific Research at the Medical University in Lodz, number RNN/726/12/KB.

#### 2.1.1. Inclusion Criteria

-Patients diagnosed with grade 2 osteoarthritis of the knee on the Kellgren and Lawrence scale;-Confirmation of OA through a standing X-ray of the knees (not older than 6 months);-No allergies to topical herbal ointment;-Absence of hypersensitivity reactions to magnetic fields and red light;-No contraindications to the therapy from other systems;-Consent to examination procedures.

#### 2.1.2. Exclusion Criteria

-Restrictions on the application of a magnetic field;-Sensory disorders;-Presence of conditions that may affect the condition of the affected joints, such as diabetes and neuroarthropathies;-Lack of patient and guardian permission for the examinations and programme participation.

### 2.2. Study Protocol—Magnetic Therapy

Patients were divided into 3 groups.

Group I—30 patients who were treated with an electromagnetic field and LED light;

Group II—30 patients who were treated with Traumeel S^®^ ointment;

Group III—30 patients who were treated with an electromagnetic field and LED light as well as with Traumeel S ointment.

The division into groups depended on the order of admission to the hospital. Treatment eligibility was assessed by the same doctor and physiotherapist.

Treatments and exercises were performed 5 times a week (15 treatment days on average). Kinesiotherapy consisted of weight-bearing and weight-bearing with resistance exercises. During a three-week stay in the hospital, on the first and last day, patients underwent clinimetric assessment using the VAS (Visual Analogue Scale—maximum 100 mm) [14] method and the Laitinen scale (maximum 16 points) [17]. Pain assessment was considered the most relevant measure for patients and in clinical assessment, as pain is the reason for seeking specialized medical help, and its remission determines the effectiveness of a given therapy. Including more results, e.g., from scales, would make the text too lengthy.

The study involved an electromagnetic field combined with light radiation emitted by LEDs generated by the Viofor JPS System from Med.&Life, Poland [18], with the additional use of Traumeel S ointment from Heel as a substance that acts as a transdermal therapeutic system (TTS). The device simultaneously emitted a variable magnetic field with an average frequency of 181.88 Hz and optical radiation in the red range, with a wavelength of 635 nm. The same test parameters were used in each patient: M3P3 (1–8); procedure time of 12 min in groups I and III.

The applicator used is an oval-shaped device that simultaneously emits a magnetic field and red light through LEDs. It has a maximum radiation power of 210 mW in a single pulse and the application area is about 20 cm^2^. It produces a magnetic field induction (peak) of 50 μT–1200 μT and a magnetic field frequency of 0.08–195 Hz. Monochromatic light is not as powerful as laser light and is eye-safe [18].

### 2.3. Data Analysis

In order to answer the research questions, statistical analyses were performed with STATISTICA StatSoft Polska 2020 (Statsoft Inc., Tulsa, OK, USA). Descriptions of clinical parameters are presented in the median (25th percentile, 75th percentile) format. Kruskall–Wallis and Dunn tests were used to compare treatment effects between the groups. The Wilcoxon test was used to compare the significance of differences in parameter values before and after treatment. The significance level was *p* = 0.05. For the variables “VAS score”, “Leitinen questionnaire score”, and “Lequesne index score”, a generalized linear model (GLM) was constructed, taking the values of a given variable before and after treatment as dependent variables and the type of treatment as qualitative factors.

## 3. Results

The results of the study are shown in the figures below, where the numerical values before and after treatment in each group are given (Figure 1, Figure 2, Figure 3 and Figure 4). A comparison of the magnitude of post-treatment pain relief between groups is also given. Table 1 shows the differences in the values of the magnitude of improvement in each group and makes a comparison between groups in this regard.

All patients evaluated together show highly significant improvement in pain intensity assessed by the VAS (Figure 1).

All patients evaluated together show highly significant improvement in pain intensity assessed by Laitinen scale (Figure 2).

In each of the studied groups, a significant difference in the intensity of pain before and after treatment is observed (Figure 3). In addition, significant differences are observed between groups I and III, I and II, and II and III, which suggest that the best results related to the reduction of pain intensity are observed in group II, followed by group I, with the highest value of pain intensity in group III. It should be noted, however, that the distribution of pain intensity at the beginning of the study was similar in the groups.

In each of the studied groups, a significant difference in the intensity of pain before and after treatment is observed (Figure 4). There were no significant differences between the groups after treatment. The last comparison of the amount of improvement in individual groups is decisive.

The comparison of the magnitude of improvement shows the greatest reduction in pain intensity in group I, then in group III, and the smallest reduction in group II, which means the greatest improvement on the VAS scale occurred after LED therapy. A similar size distribution is observed on the Laitinen scale, except that there are no statistically significant results here. However, the results of group III do not confirm the assumptions about the synergism of combined transdermal therapy.

No adverse symptoms were observed in patients in any of the study groups.

## 4. Discussion

The purpose of this study was to evaluate and compare the effectiveness of an electromagnetic field combined with light radiation emitted by LEDs and Traumeel S ointment on physiotherapy results in patients with knee osteoarthritis, as measured by the VAS and Laitinen scale. Patients with radiological grade 4 OA on the Kellgren and Lawrence scale were excluded from the study. The severity of the lesions did not respond to any of the conservative treatment methods. Patients with such severe changes were referred for surgical treatment. Comparing all groups of patients included in the study, we can conclude that each treatment method significantly changed the values of the parameters in both scales used. The group of patients with the greatest positive results in terms of pain reduction was the group of patients that received ELF-EMF and LED light treatment. The second-greatest pain reduction was experienced by the group of patients who received ELF-EMF and LED light treatment as well as Traumeel S ointment.

With respect to the electromagnetic field and LED light treatment, many authors in their articles have confirmed the analgesic effectiveness of this treatment. ELF-EMF and LED light is a treatment using optical radiation from the visible and infrared light spectrum, which is produced by high-energy LEDs (light-emitting diodes). LEDs emit light radiation in the red (R), infrared (IR), as well as mixed (RIR) light spectrum that is distinguished by its monochromaticity and incoherence. The biological effect of LED therapy is mainly related to the effect of heat on tissues. This causes, among other things, dilation of skin capillaries, increased metabolism and healing processes, reduction in skeletal muscle tension and, importantly, an increase in one’s pain threshold. The body’s response to IR can be local or global, depending on the amount of energy absorbed. Monochromatic light applicators can emit radiation with wavelengths similar to those of laser radiation, but without coherence and polarization effects. The density and energy of the light beam are high enough to induce photostimulation, but without the danger of tissue damage that laser radiation poses, as well as other side effects associated with high-energy wavelengths. Thus, LED therapy, in terms of therapeutic doses, replaces laser therapy with a scanner, without having to comply with health and safety regulations required for the use of lasers [11,14,19,20]. In ELF-EMF and LED light treatment, an important physical agent is an alternating low-frequency magnetic field ranging from a few to 3000 Hz, with induction ranging from 1 pT to 100 μT. An important detail of the action of magnetic fields is the increase in the secretion of endogenous opiates of the β-endorphin group, which exhibit analgesic properties by acting on the central nervous system. The simultaneous application of these two therapies results in a synergistic effect, which is helpful in diseases of the osteoarticular, muscular and nervous systems [11,14,19,21,22].

The second important element in the study was the use of Traumeel S ointment in patients with OA. This drug works with plant and mineral ingredients in a way that is different from analgesics and non-steroidal anti-inflammatory drugs. A meta-analysis of the data showed that natural products inhibit the expression of syndecan IV, MMP1, MMP3, MMP19, ADAMTS-4, ADAMTS-5, iNOS, COX-2, collagenases, TNF-α, IL-1β and IL-6 in vitro and in vivo. Cytokines also increase the expression of collagen II and aggrecan. Extracts and isolated compounds that affect major signaling pathways include SIRT, MAPK, AMPK, NLRP3, PI3K/AKT, mTOR, NF-κB, WNT/β-catenin, NRF2 and JAK/STAT3. They also affect modes of cell death such as apoptosis, autophagy, pyroptosis and ferroptosis. The plants from which these medicines are derived are used in the production of Traumeel S ointment and gel. The ointment acts supportively after injuries such as joint sprains, minor injuries (e.g., contusions, hematomas, bruises), and in cases of muscle and joint pain. It relieves pain and swelling and reduces inflammation in traumatic injuries [9,10]. Many authors have confirmed the effectiveness and good tolerability of the use of Traumeel S ointment in their works [16,23,24]. Non-steroidal anti-inflammatory drugs (NSAIDs) have become the primary drugs used to treat pain, trauma and inflammation [25,26]. A therapeutic alternative to NSAIDs in the treatment of OA is Traumeel S. Non-steroidal anti-inflammatory drugs (NSAIDs) and Traumeel S ointment have shown comparable analgesic efficacy in patients with OA [27,28]. The composition of Traumeel S ointment was developed in a targeted manner. It was guided by a synergistic, regulatory effect on individual symptoms of inflammation. Traumeel S has a mechanism of action that is different from that of NSAIDs, acting through complex interactions in the system of anti-inflammatory and pro-inflammatory markers, thus regulating the inflammatory process and shortening its duration. The ointment has analgesic, antiedematous and anti-inflammatory effects. The formulation’s ingredients have been selected in a way that takes into account various aspects of the inflammatory process in the musculoskeletal organ, which accompanies any injury. Traumeel S is an effective drug in the treatment of sports injuries and is used by high-performance athletes as well as by amateur athletes. The drug addresses the causes of muscle pain and can be used in long-term therapy. Traumeel S, due to its mechanism of action, is used to treat inflammation in general, not only that resulting from injury. Therefore, the drug is recommended for patients with chronic pain in the treatment of inflammatory and degenerative diseases of the musculoskeletal system [28]. The composition of the ointment also allows it to cover areas where there are abrasions of the epidermis and open wounds, which certainly accelerates the healing process. Admittedly, this study did not confirm the synergistic effect of TTS, but nevertheless the analgesic results were satisfactory. Besides the analgesic properties, other aspects of the synergistic effect of the two therapies were not investigated in this study. Perhaps, however, it would have turned out that synergism occurs in the healing of open injuries. A shortcoming of the study is that it did not test Traumeel S gel instead of ointment. Unfortunately, the Heel company allowed us to test the ointment without giving us the opportunity to test the gel simultaneously in another study group.

Nevertheless, the study shows an improvement in pain when the ointment is used as a sole analgesic, which has already been discussed in some papers [16,27], and the ointment may be an alternative to topically applied non-steroidal anti-inflammatory agents. In addition, the use of a magnetic field along with LED light has been shown to be an effective pain reliever. It should be remembered that pain is a direct result of inflammation and, as it should be assumed, the analgesic effect is combined with the anti-inflammatory effect. Thus, this form of treatment may be an opportunity to replace pharmacotherapy, along with its side effects, especially considering that it is characterized by the formation of the phenomenon of biological hysteresis.

A limitation of this study is the subjective assessment of pain, although it was expressed in two scales. It is advisable to objectify the data in the study of the patients’ condition using functional scales and muscle strength tests, which is planned for a future publication of the continuation of this study. It is also worth conducting a follow-up examination 30 days after the end of the examination, bearing in mind the effect of biological hysteresis that occurs after magnetophoresis [13]. A shortcoming of this study is the lack of a comparison group, which was not included for ethical reasons. In such a case, patients would be deprived of a significant part of therapy. Since the ointment form probably appears to be a magnetic field inhibitor and a light absorber, it would be worthwhile to carry out studies to investigate similar groups using Traumeel S^®^ gel instead.

## 5. Conclusions

The applied therapy showed that treatment involving an electromagnetic field combined with light radiation emitted by LEDs and the use of Traumeel S ointment gives positive results in reducing pain. Whether they are used individually or together, the indicated elements of the therapy provide significant pain reduction. The most potent analgesic factor seems to be ELF-EMF and LED light used separately. Traumeel S in magnetoledophoresis does not act synergistically with the magnetic field, which is probably due to the type of substrate applied in the ointment and the type of organic, plant-derived ingredients used. It would be advisable to conduct similar studies with a preparation that has a different base, i.e., with the Traumeel S gel produced by the company. Traumeel S ointment, on the other hand, may be an alternative to topical NSAIDs.

## Figures and Tables

**Figure 1 ijerph-20-03696-f001:**
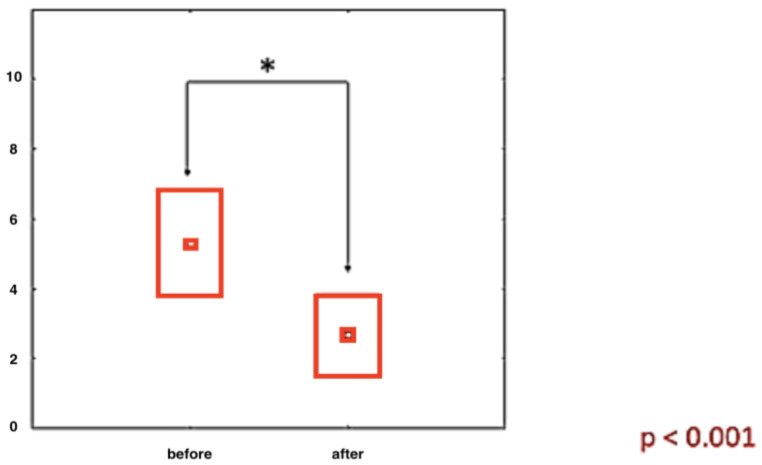
The magnitude of VAS pain intensity in patients from all groups collected together before and after treatment (results in cm). *—statistically significant.

**Figure 2 ijerph-20-03696-f002:**
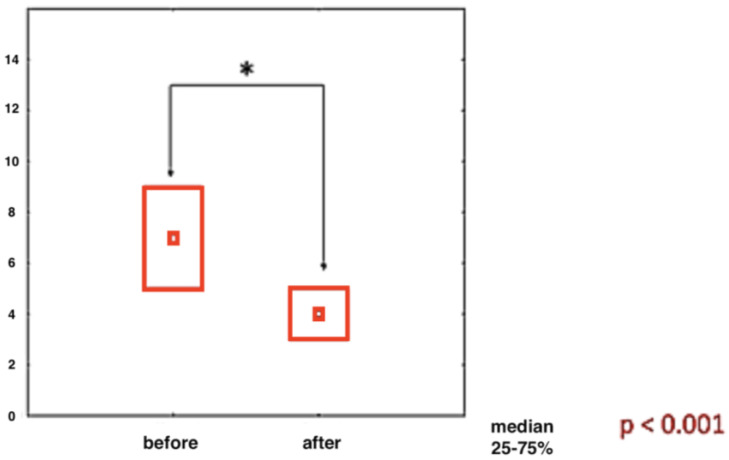
The magnitude of pain intensity on the Laitinen Scale in patients from all groups collected together before and after treatment (results in points). *—statistically significant.

**Figure 3 ijerph-20-03696-f003:**
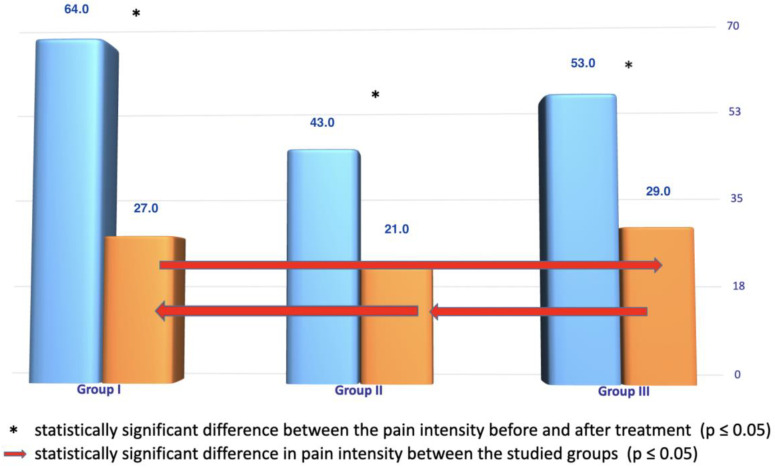
Comparison of pain levels on the VAS before and after therapy in each group and between groups after therapy (results in mm).

**Figure 4 ijerph-20-03696-f004:**
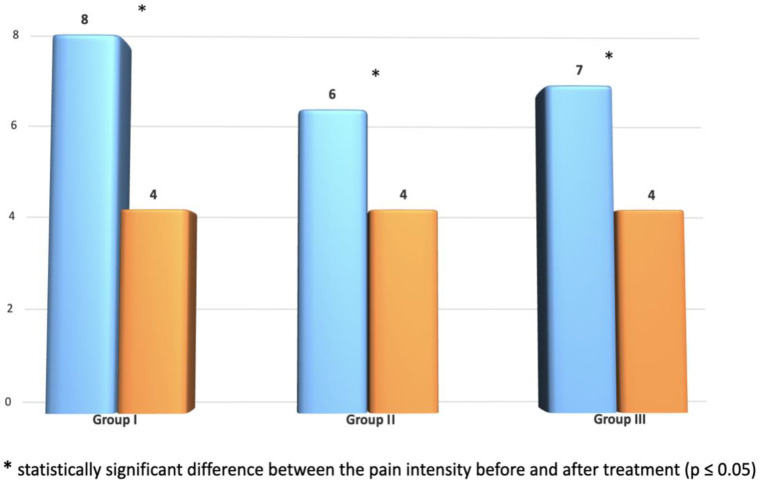
Comparison of the level of pain on the Laitinen scale before and after therapy in each group and between groups after therapy (results in points).

**Table 1 ijerph-20-03696-t001:** The differences (magnitude of improvement) between groups before and after treatment.

Parameter	Test K-W (*p*)	Group I	Group II	Group III
Difference on the VAS	*p* = 0.003	35.5	18.5	26.5
Difference on the Laitinen questionnaire	*p* = 0.286	3.0	2.0	2.5

## Data Availability

Not applicable.

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
