# Peer review of "Analgesic Effectiveness of Physical Therapy Combining the Use of Electromagnetic Fields with Light Radiation Emitted by LEDs along with the Use of Topical Herbal Ointment in Patients with Gonarthrosis"

_ijerph, 2023, doi:10.3390/ijerph20043696_

Round 1
Reviewer 1 Report
It is a novel work, in a pathology little investigated, with an innovative approach and good methodological design.
The abstract is brief, concrete and structured. However, it does not show objective results or numerical evaluations of the results or differences between treatments. This must be included.
The introduction is complete, well-founded and raises the necessary theoretical concepts and hypotheses and objectives.
The methodology is good, well described and provides sufficient detail to ensure reproducibility.
It is interesting that the use of 3 therapeutic groups should be used for a concordant analysis.
The results are clear, concrete, although little described. The measurement approach in a pathology with subjective components is adequate and of high quality. Measurements are expressed correctly and objectively. However, given the numerical and baseline differences between groups, the graphical expression of proportional percentage variations in each group would be good.
The discussion is adequate, it is necessary to complete it with a better description of the limitations and future applications of this work.
It is interesting that the data seems to show a better result with treatment 1 than with treatment 3 (which is a sum of 1 and 2). Therefore, there are elements to suspect that group 2 therapy is not only less effective, but also that it seems to reduce the efficacy of group 1 treatment when combined. For this reason, the conclusion does not seem to be appropriate, since it suggests a sort of similar effectiveness in all therapeutic strategies when in reality the design is intended for another type of analysis. Said analysis provides clear data that shows the non-equality of the treatments.
Author Response
Dear Reviewer,
thank you very much for all valuable comments.
We've improved the article according to your suggestions.
It is a novel work, in a pathology little investigated, with an innovative approach and good methodological design.
Thank you very much.
The abstract is brief, concrete and structured. However, it does not show objective results or numerical evaluations of the results or differences between treatments. This must be included.
It was corrected.
The introduction is complete, well-founded and raises the necessary theoretical concepts and hypotheses and objectives.
Thank you very much.
The methodology is good, well described and provides sufficient detail to ensure reproducibility.
Thank you very much.
It is interesting that the use of 3 therapeutic groups should be used for a concordant analysis.
Thank you very much.
The results are clear, concrete, although little described. The measurement approach in a pathology with subjective components is adequate and of high quality. Measurements are expressed correctly and objectively. However, given the numerical and baseline differences between groups, the graphical expression of proportional percentage variations in each group would be good.
Thank you very much. It would be difficult at the moment, but if it is necessary, we will correct it.
The discussion is adequate, it is necessary to complete it with a better description of the limitations and future applications of this work.
Thank you very much. The section on limitations has been updated.
It is interesting that the data seems to show a better result with treatment 1 than with treatment 3 (which is a sum of 1 and 2). Therefore, there are elements to suspect that group 2 therapy is not only less effective, but also that it seems to reduce the efficacy of group 1 treatment when combined. For this reason, the conclusion does not seem to be appropriate, since it suggests a sort of similar effectiveness in all therapeutic strategies when in reality the design is intended for another type of analysis. Said analysis provides clear data that shows the non-equality of the treatments.
Thank you very much for your constructive analysis. Conclusions have been corrected.
Reviewer 2 Report
This study evaluated the effectiveness of physical therapy combining electric and light radiation with a topical ointment called Traumeel S for reducing pain in people with knee osteoarthritis. Ninety participants were divided into three groups: Group I received electric and light therapy with Traummeel S ointment, Group II received only Traumeel S ointment, and Group III received electric and light treatment with the ointment. The researchers used a Visual Analogue Scale (VAS) and the Laitinen Scale to measure the participants' pain levels before and after treatment. The results showed that combining electric and light therapy with Traumeel S ointment significantly reduced pain levels.
The paper has comprehensive and considers the relevant literature regarding the analgesic effects of this treatment. The discussion of the Traumeel S ointment is also complete, considering the data from meta-analyses and its impact on major signaling pathways. However, I suggest more detail be provided on the use of the Traumaal S ointment, including the dosage administered to the patients and how it was administered, as well as any potential side effects or adverse reactions. Additionally, providing possible explanations, such as mechanisms. The ingredients used in Traumeel S ointment include a combination of homeopathic-grade plant extracts, such as arnica montana, belladonna, chamomile, echinacea purpurea, and hypericum perforatum, among others. These substances act synergistically to help reduce inflammation and pain, enhance the body's natural healing response, and provide a soothing effect to affected tissues. Additionally, studies have shown that Traumeel S ointment is comparable to non-steroidal anti-inflammatory drugs (NSAIDs) in terms of analgesic efficacy in patients with osteoarthritis. Further research is needed to confirm the full potential of Traumeel S for treating sports-related injuries and musculoskeletal conditions.
Here are my comments to help get this manuscript amended:
1. The sample size was relatively small, which could limit the external validity of the results.
2. No control group was included in the study, making it difficult to determine the effects of each individual treatment or the combination of treatments on pain reduction.
3. The study population and sample size should be more clearly defined, the inclusion and exclusion criteria should be specified with greater detail and clarity, the type of magnetic field and LED light used should be provided, and
4. The data analysis techniques should be described more fully as not sure why Kruskal–Wallis test and the Wilcoxon signed-rank test were especially sued.
5. The methodology used to measure the outcomes (pain) should be more detailed and explain why pain is more important than other outcomes, such as function and quality of life.
6. The data analysis process should be described in more detail and used tables with values of the outcome after controlling for covariate factors.
7. There should be more clarity on the magnitude of improvement in individual groups and a more thorough comparison of the magnitudes of improvement across the three groups. Adverse symptoms should also be discussed in more detail.
8. The potential limitations of the study should be discussed.
9. The study's findings should be compared with prior literature to support its conclusions further.
Author Response
Dear Reviewer,
thank you very much for all valuable comments.
We've improved the article according to your suggestions.
1.The sample size was relatively small, which could limit the external validity of the results.
Despite the prevalence of gonarthrosis, gathering the group of patients described in the study was quite difficult and took several years. A large number of patients were excluded from the study for the reasons listed in the conditions of exclusion. Also the reasons for inclusion limited the size of the study groups.
2.No control group was included in the study, making it difficult to determine the effects of each individual treatment or the combination of treatments on pain reduction.
Good point. A study with a control group and a larger number of patients is planned in the future. We've added this information to the limitations section.
However, it is quite difficult in the case of creating a control group, because it is impossible to avoid abandoning the full form of treatment and rehabilitation in this case.
3.The study population and sample size should be more clearly defined, the inclusion and exclusion criteria should be specified with greater detail and clarity, the type of magnetic field and LED light used should be provided, and
An insightful note. The comments have been taken into account in the text. Broader descriptions regarding the applied magnetic field and LED light are in the discussion.
4.The data analysis techniques should be described more fully as not sure why Kruskal–Wallis test and the Wilcoxon signed-rank test were especially sued.
The comments have been taken into account in the text.
5.The methodology used to measure the outcomes (pain) should be more detailed and explain why pain is more important than other outcomes, such as function and quality of life.
The pain score was considered the most relevant to patients and in clinical assessment, as pain is the reason for seeking specialized medical help, and its resolution determines the effectiveness of a given therapy. The inclusion of more results, such as from scales, would have exceeded the volume of the presented text.
6.The data analysis process should be described in more detail and used tables with values of the outcome after controlling for covariate factors.
Values in the VAS and Laitinen scales are given in the figures and in the table.
7.There should be more clarity on the magnitude of improvement in individual groups and a more thorough comparison of the magnitudes of improvement across the three groups. Adverse symptoms should also be discussed in more detail.
Since no side effects were observed, their possible symptoms were not described in detail. Possible side effects of the ointment are included in the text, although they were not found in any case.
8.The potential limitations of the study should be discussed.
The section on limitations has been updated.
9.The study's findings should be compared with prior literature to support its conclusions further.
The study was made as completely original and possible comparisons with the magnetic field effect in magnetophoresis and the ointment used were made. However, references to magnetoledophoresis have not been found anywhere in the literature.
Round 2
Reviewer 1 Report
The authors have made the requested changes. Some additional changes are necessary that correspond to style issues (the citation of tables and graphs in the text is missing, which can be deduced based on its content, but has been omitted by the authors). Apart from what has been mentioned, there are no major drawbacks, and I believe that it can be accepted for publication unless another reviewer opines otherwise.
Author Response
Dear Reviewer,
thank you for all your valuable comments.
We have made changes as you suggest.
Best regards